# Microtomographic investigation of a large corpus of cichlids

**David Haberthür** [1]*, **Mikki Law**[2,3], **Kassandra Ford**[2,3,4], **Marcel Häsler**[2,3], **Ole Seehausen** [2,3], **Ruslan Hlushchuk**[1]

**1** Institute of Anatomy, University of Bern, Bern, Switzerland, **2** Aquatic Ecology and Evolution, Institute of Ecology and Evolution, University of Bern, Bern, Switzerland, **3** Department of Fish Ecology and Evolution, Eawag, Swiss Federal Institute for Aquatic Science and Technology, Kastanienbaum, Switzerland, **4** Department of Biological Sciences, George Washington University, Washington, DC, United States of America

* david.haberthuer@unibe.ch

**Data Availability Statement:** All relevant data (log files and data for generating the figure of the otolith) are within the manuscript and its Supporting information files, they are linked from

## Abstract

A large collection of cichlids (N = 133) from Lake Victoria in Africa, with total lengths ranging from 6 to 18 cm was nondestructively imaged using micro-computed tomography. We present a method to efficiently obtain three-dimensional tomographic datasets of the oral and pharyngeal jaws and the whole skull of these fishes to accurately describe their morphology. The tomographic data we acquired (9.8 TB of projection images) yielded 1.5 TB of three-dimensional image stacks used for extracting the relevant features of interest. Herein we present our method and outlooks on analyzing the acquired data; a morphological description of the oral and pharyngeal jaws, a three-dimensional geometric morphometrics analysis of landmarked skull features, and a robust method to automatically extract otoliths from the tomographic data.

## Introduction

### History and rationale

Cichlid fish in African lakes are powerful model systems for evolutionary biology in speciation and adaptive evolutionary radiation research [1–3]. The functional decoupling of their oral and pharyngeal jaws is hypothesized to be a factor in making cichlids unusually versatile in their feeding strategies, allowing them the ability to adapt to a wide range of environmental factors [4]. The hypothesis is that the fusion of the lower pharyngeal jaws enables it to become an adaptable tool for processing food, in turn releasing the oral jaws from functional constraint. The oral jaws no longer need to process prey and can therefore specialize on prey capture.

Even though the evolutionary diversification of cichlid fish radiations in Lake Victoria, East Africa is a well researched topic, it remains a complex system in need of further study [2, 5]. We aim to better understand the functional anatomy of the skulls and jaws in these fish in order to test the functional decoupling and other hypotheses about what may facilitate exceptionally high rates of morphological evolution.

The collection of Lake Victoria cichlids investigated here is extremely valuable, hence a nondestructive imaging method for studying these samples is paramount.

the text. The data for the otolith is downloaded within the Jupyter notebook. We cannot provide all data for all the fish, as this is more than 10 TB of images.

**Funding:** The authors received no specific funding for this work.

**Competing interests:** The authors have declared that no competing interests exist.

Since micro–computed tomography can be regarded as a nondestructive investigation method for biological samples, it is the ideal imaging method for investigating the oral and pharyngeal jaws as well as the skull features of the fish presented in this study [6] (N = 133).

## Micro-computed tomography

X-ray microtomography is a valuable tool to gain insights into the inner structure of highly diverse samples, namely for specimens related to the biomedical sciences [7]. Microtomographic imaging has been employed as a method of choice to nondestructively assess the morphology of different fish species, large and small [6]. For a small overview of analyses which are possible with X-ray microtomographic imaging in relation to fish biology and morphology, see prior work of the authors of this manuscript [6, 8, 9] as well as other studies [10].

Depending on the structures of interest to be assessed, biomedical samples are often only tomographically scanned after them being stained with a contrast agent [11], most often applying contrast agents that contain heavy metals [12]. This is done to increase the contrast difference between different structures in the sample. Due to their inherent contrast difference to the surrounding tissue, the structures of interest we touch upon in this manuscript (teeth and bones, i.e. jaws and skull) are well visualized in the unstained samples. Thus, staining the samples presented in this manuscript prior to imaging was not necessary.

## Materials, methods and results

### Sample procurement and preparation

The fish were kept in 75% ethanol for long-term storage at the Swiss Federal Institute of Aquatic Science and Technology (Eawag). They were delivered to the Institute of Anatomy for micro-CT imaging sorted into batches of approximately equal length.

### Ethics statement

The study used specimens from an already existing fish collection built over decades of research led by Ole Seehausen that continues to this day. All relevant permits for research, exports from the respective country and import to Switzerland were given and the research has followed modern ethical guidelines for scientific research.

### Micro-computed tomographic imaging

All samples were scanned on two of the three available high-resolution micro-CT machines of the Institute of Anatomy of the University of Bern in Switzerland, a SkyScan 1272 and a Sky-Scan 2214 (both Bruker microCT, Kontich, Belgium).

The fish were sorted into 'bins' based on their physical size. We used a custom-made sample-holder to scan each of the fish in our machines. It was 3D-printed on a Form 2 desktop stereolithography printer (Formlabs, Somerville, Massachusetts, USA), the file for printing the holder is available online [13] as part of a library of sample holders for tomographic scanning of biomedical samples [14]. The original *OpenSCAD* [15] file [16] is parametrized to effortlessly generate a file for 3D-printing sample holders to accommodate the varying width, height and length classes of the fish.

In total, we acquired 372 tomographic scans of 133 different specimens. All the scanning parameters are collected in a table in the Supporting information; an overview of the parameters is given below.

Since the fish varied greatly in their length (total length varied between 6 cm and 18 cm), the voxel size of each dataset also varied greatly. We acquired datasets with (isometric)

voxel sizes ranging from 3.5 to 50 μm. Due to the geometric magnification employed by the Bruker SkyScan microCT machines, the voxel size of the resulting dataset is dependent on the position of the specimen in the machine. Naturally, the field of view of the resulting dataset is also dependent on the chosen voxel size. The chosen voxel size is thus scan-specific. The voxel size was chosen in a way that the region of interest would fit into the available lateral field of view of the microCT machine, as well as having the smallest required voxel size that could resolve the structures of interest. Since the region of interest (either oral or pharyngeal jaws or the complete skulls) often did not fit into a single field of view *along* the anteroposterior axis of the fish, we often performed several tomographic scans along the rotation axis of the specimen in the machine (since the specimen were all scanned rotating around the anteroposterior axis). The projection images from those so-called stacked scans are then automatically merged to one stack of reconstruction images by the reconstruction software.

Depending on the size of the specimen we set the X-ray source voltage to 50–80 kV and—depending on the voltage—to a current between 107 and 200 μA. The different voltages and currents were chosen according to the Bruker guidelines for ideal image acquisition. The X-ray spectrum was filtered either by an aluminum filter of varying thickness (either 0.25, 0.5 or 1 mm for increasing specimen size) before digitization to projection images or recorded in an unfiltered way (for smaller specimen). Projection images spanning either 180˚ or 360˚ of sample rotation were acquired in angular steps of 0.1˚, 0.15˚ or 0.2˚, also depending on size of the fish. In total we recorded 9.8 TB of projection images (TIFF and *.iif files, where the *.iif files are for the so-called alignment scans).

All the recorded projection images were subsequently reconstructed into three-dimensional stacks of axial PNG images spanning the desired regions of interest of each specimen. All the specimens were scanned with their mouths facing downward in the sample holder and rotating along their long axis. We manually aligned each of the reconstructed datasets so that the lateral axis of the fish was horizontal in relation to the x and y direction of the reconstructed slices. We reconstructed the projection images with NRecon (Version 1.7.4.6, Bruker microCT, Kontich, Belgium) with varying ring artifact and beam hardening correction values, depending on each fish (again, all relevant values are listed in the Supporting information). In total, this resulted in 1.5 TB of reconstruction images (a bit more than one million *rec*.png files). On average, each of the scans we performed is made up of about 2700 reconstruction images.

A small script [17] was used to generate redundant (archival) copies of the raw projection images and copy all the files to a shared network drive on the 'Research Storage' infrastructure of the University of Bern, enabling collaboration on the data by all authorized persons at the same time. This automated archival and copying process made it possible to delete the raw projection images on the production system shortly after acquisition, freeing resources and easing data handling. A subset of the data, namely the log files and reconstructed image stacks were always kept on the production system for image processing (as described in Preparation for analysis below).

## Data analysis

We wrote a set of *Jupyter* [18] notebooks with *Python* code to work with the acquired images and to extract the desired data. The notebooks were written at the start of the project, to be able to process new scans as soon as they were reconstructed. Re-runs of the notebook added newly scanned and reconstructed data to the analysis, facilitating an efficient quality check of the scans and batched processing of the data.

All notebooks written for this work are available online [19] and are extensively commented. The modularized and interactive format chosen for these notebooks makes them easy to adapt for any other study dealing with tomographic data. The first author of this study is happy to help other scientists to adapt the notebooks to their use-case. The notebooks are split into different tasks performed for this study and are specifically mentioned in the respective sections below.

**Preparation for analysis.** The main *Jupyter* notebook for this manuscript dealt with reading all log and image files and preparing overview images of each scan for quality checking and further analysis.

At first, *all* log files of *all* the scans present in the processed folder were read into a *pandas* [20, 21] dataframe. This already enabled us to extract the specimen name and scan. Since we performed multiple scans for each specimen, i.e. a low resolution scan with large field of view for the whole head and one or two scans in high resolution focusing on the region of the oral and pharyngeal jaws, it was necessary to be able to efficiently find the scan to be looked at in detail. From the log files we extracted the scanning and reconstruction parameters of each performed scan to double-check them. This allowed us to exclude scans with unexpected values or errors and correct for those prior to a next run of the notebook. All relevant values for each scan were also saved into the aforementioned dataframe. This allows for an easy collation of all the relevant values into a table (as shown in the Supporting information) at the end of each run of the notebook.

After displaying several parameters of both data acquisition and reconstruction for ruling out any operator error we used *Dask* [22] to efficiently access the tomographic data on disk (in the end amounting to a total of nearly a million single images). On average, each of the tomographic datasets contain around 2700 slices, so the total size of the acquired data (1.5 TB) exceeds the RAM size available on an average high-end workstation. The use of *Dask* (and more specifically *dask-image* [23]) facilitated on-demand loading of the needed data directly from disk for each of the specimen to be analyzed in each step of the notebooks.

At first, we extracted the central view of each of the three axial directions of the datasets (i.e. 'anteroposterior', 'lateral' and 'dorsoventral' view) and either saved those to disk or loaded them from disk if they were already generated in prior runs of the notebook. The notebook then also generated the maximum intensity projection (MIP) for each of the anatomical planes and either saved them to disk or loaded them from prior runs.

At the end of the notebook we performed a final sanity check on the MIP images. In this check we examined the mapping of the gray values of the raw projection images to gray values in the reconstructions, i.e. checked that no overexposed pixels were present. This is an efficient way to double-check the gray value mapping, since the MIP images have already been generated in prior steps of the notebook and contain the highest gray values present in all the reconstructed images for each scan.

## Image processing

**Extraction of oral and pharyngeal jaws, visualization of tomographic data.** To extract the oral jaw (OJ) and pharyngeal jaw (PJ) of the fish, we used *3D Slicer* (Version 4.11.20210226) [24] extended with the *SlicerMorph* tools [25] which aim to help biologists work with 3D specimen data. The reconstructed image stacks were loaded into *ImageStacks*, depending on their size we reduced the image resolution (i.e. downscaled the images) for this first step. The three-dimensional volume was rendered via *VTK GPU Ray Casting*. A custom-made volume property was used as an input to view the scans. Using toggles in the volume rendering, we defined regions of interest (ROIs) for both the OJs and PJs in each specimen. These

ROIs were then extracted in their native resolution from the original dataset for further processing. Using the gray value thresholding function in *3D Slicer*'s *Segment Editor*, the teeth in both the oral and pharyngeal jaws were extracted. We used the *Scissor* and *Island* tools of the *Segment Editor* to isolate single regions.

Processed regions of interest were then exported as Nrrd [26] files. The three-dimensional visualizations of all regions of interest were compiled into overview images (see Fig 1 for an example from the compilation document). In total, we compiled such overview images for 133 specimens with full head morphology, oral jaw and lower pharyngeal jaw profiles.

**Principal components analysis of skull landmarks.** Current studies are using 3D geometric morphometrics to compare the morphological shape of these scanned cichlids using statistical analysis. We used a homologous landmark scheme across one-half of the skull for higher density of shape information [9, 28], and landmarks were placed on each specimen using *3D Slicer*. To examine differences in shape across the species sampled, we performed a Generalized Procrustes Superimposition on the landmark data to remove the effects of location, size, and rotation from the analysis using the *geomorph* package in *R* (Version 4.2.1) with

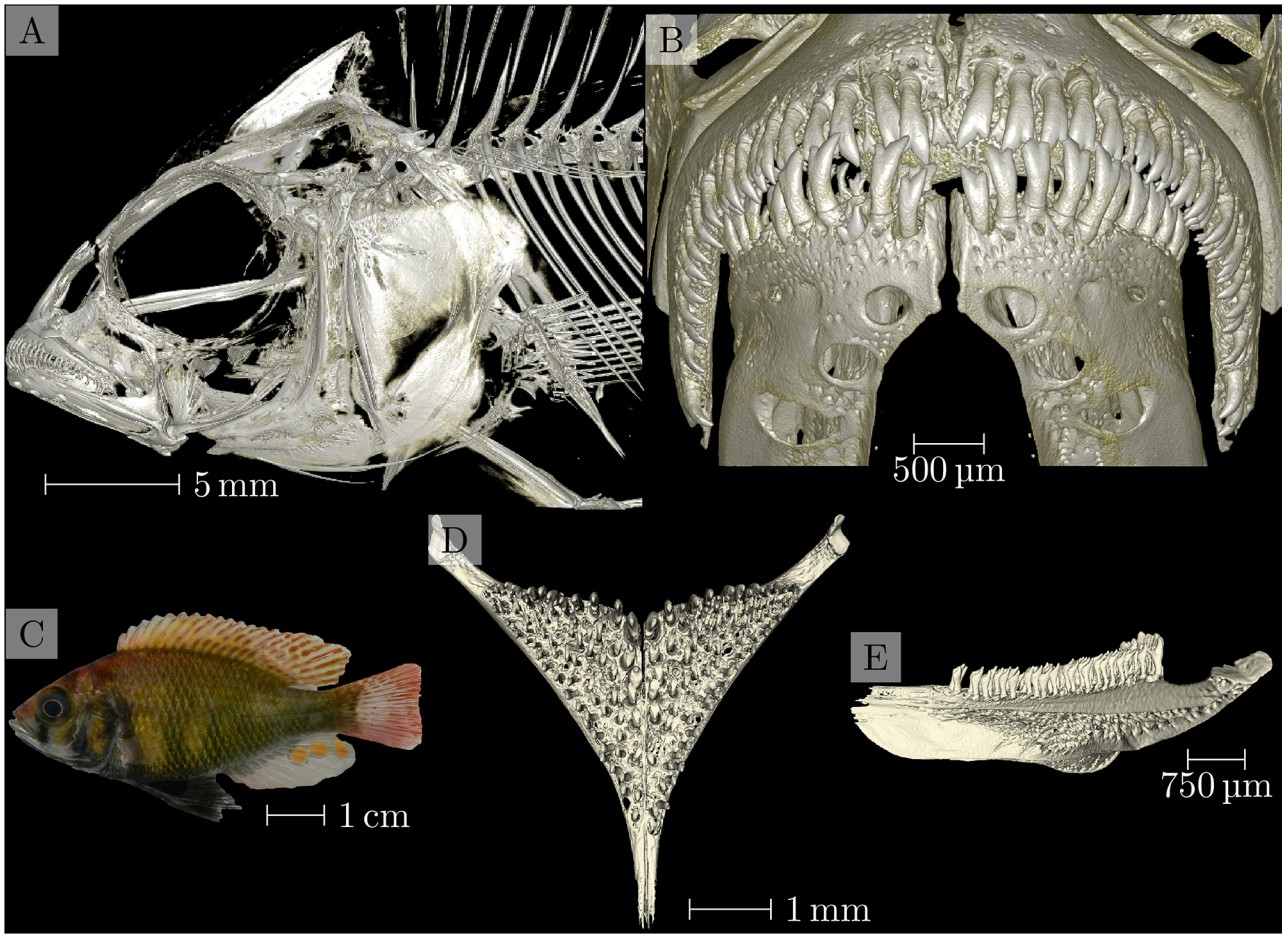

**Fig 1. Overview of data from sample 104016, enterochromis I cinctus, from 'Station E' in the transect of Mwanza Bay at the southern edge of Lake Victoria.** A map showing the detailed location of 'Station E' is shown in Fig 1 of [27]. Panel A: Three-dimensional visualization of the head scan. Panel B: Three-dimensional visualization of the oral jaw scan. Panel C: Photograph of the specimen. Panels D and E: Three-dimensional visualization of the pharyngeal jaw, dorsoventral and lateral view, respectively.

*RStudio* (Version 2022.07.2+576) [29–33]. This process brings all specimens to a common origin, scales the landmarks to a unit centroid size, and rotates specimens to reduce distances between landmarks. A principal components analysis was then performed in *geomorph* on the superimposed landmark data to visualize the major axes of shape change across sampled species. We then used phylogenetic information to identify instances of repeated evolution of trophic adaptations in these cichlids.

**Automatic extraction of otoliths.** Otoliths are structures made up of mostly calcium carbonate located in the head of fishes. Due to their composition, they are easily distinguished in the X-ray images we acquired. We devised an image processing method to automatically and robustly detect the location of the otoliths in the tomographic scans of the skulls of the cichlids. Based on this detected location we were then able to efficiently extract a cropped region of interest from the original data which includes only the region of the otoliths. The whole method is implemented in its own *Jupyter* notebook (part of the aforementioned analysis repository [19]).

Since we took great care to scan the fish parallel to their anteroposterior direction and reconstructed the tomographic datasets parallel to the lateral and dorsoventral direction of the fish, we could use this 'preparation' for automatically extracting the otoliths. By extracting both the peaks and the peak widths of the gray values along both the horizontal and vertical direction of the MIP (generated above) we robustly detect the position of the otoliths in the datasets. The robust detection is supported by suppressing a small, configurable part of each region, i.e. the front and back, top and bottom or the flanks.

Fig 2 shows the visualization of the process. The colored horizontal and vertical bars in each of the directional MIPs denote the found peak location of the two values found in the two

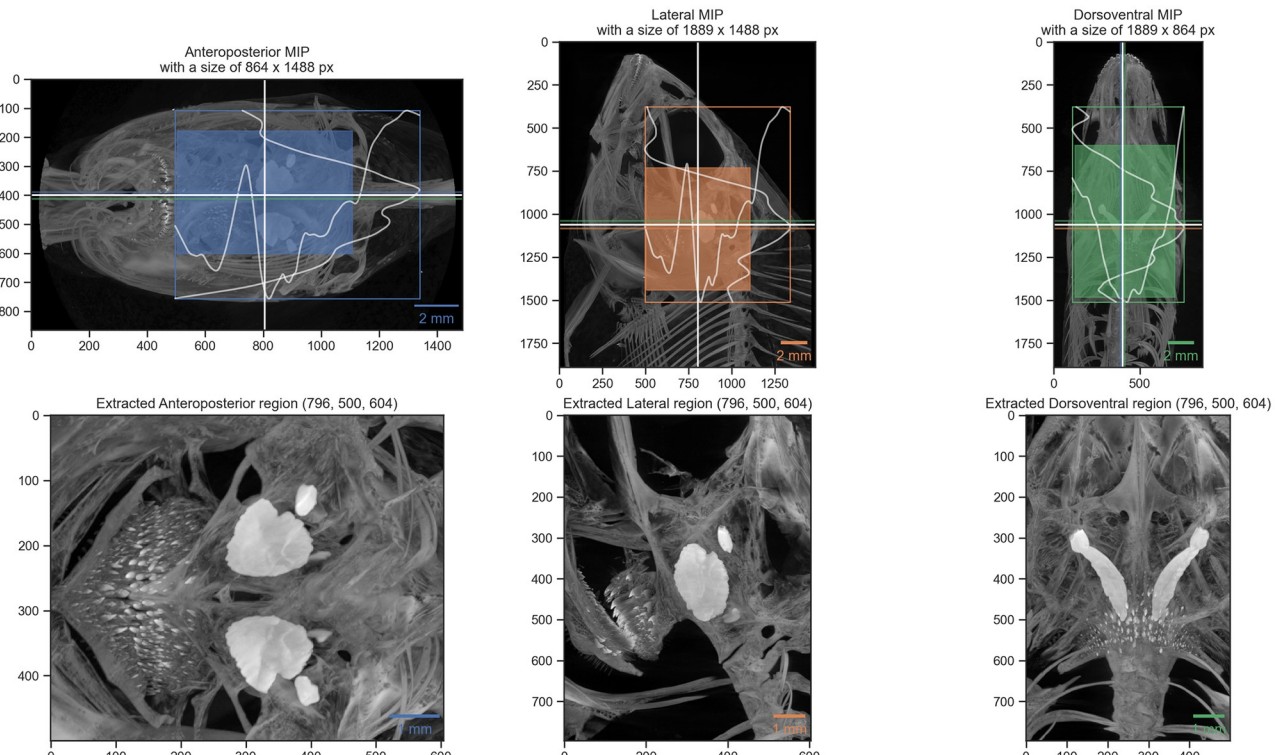

**Fig 2. Visualization of automatic otolith extraction.** The top row shows the found location of the otolith center in each of the three anatomical directions. The bottom row shows a maximum intensity projection of each of the three anatomical directions of the extracted otolith region.

other directional views. The white bars show the mean of these two detected positions, which were used for extracting the otoliths from the original datasets. Making use of the *Dask* library facilitated efficient access to all the data on disk and writing out small, cropped copies of the datasets around the otolith positions.

By detecting the largest components in the cropped copies of the datasets we can easily extract and visualize the otoliths in 3D, as shown in Fig 3. The extracted otoliths are thus prepared for further analysis and visualization. The simple three-dimensional visualization is integrated in the aforementioned *Jupyter* notebook through an integrated visualization library [34] and is also shown in the Supporting information.

The notebook for extracting the otoliths can be run in your browser without installing any additional software (via *Binder* [35]). To do this, one starts the notebook by clicking a single button in the README file of the project repository [19]. This starts a computing environment in the cloud, downloads the tomographic data we acquired of *one* specimen, and performs both the otolith extraction and visualization in your browser.

## Discussion

We acquired high resolution tomographic datasets of a large collection of cichlids (N = 133), several tomographic scans were performed for each specimen. The 372 acquired datasets were

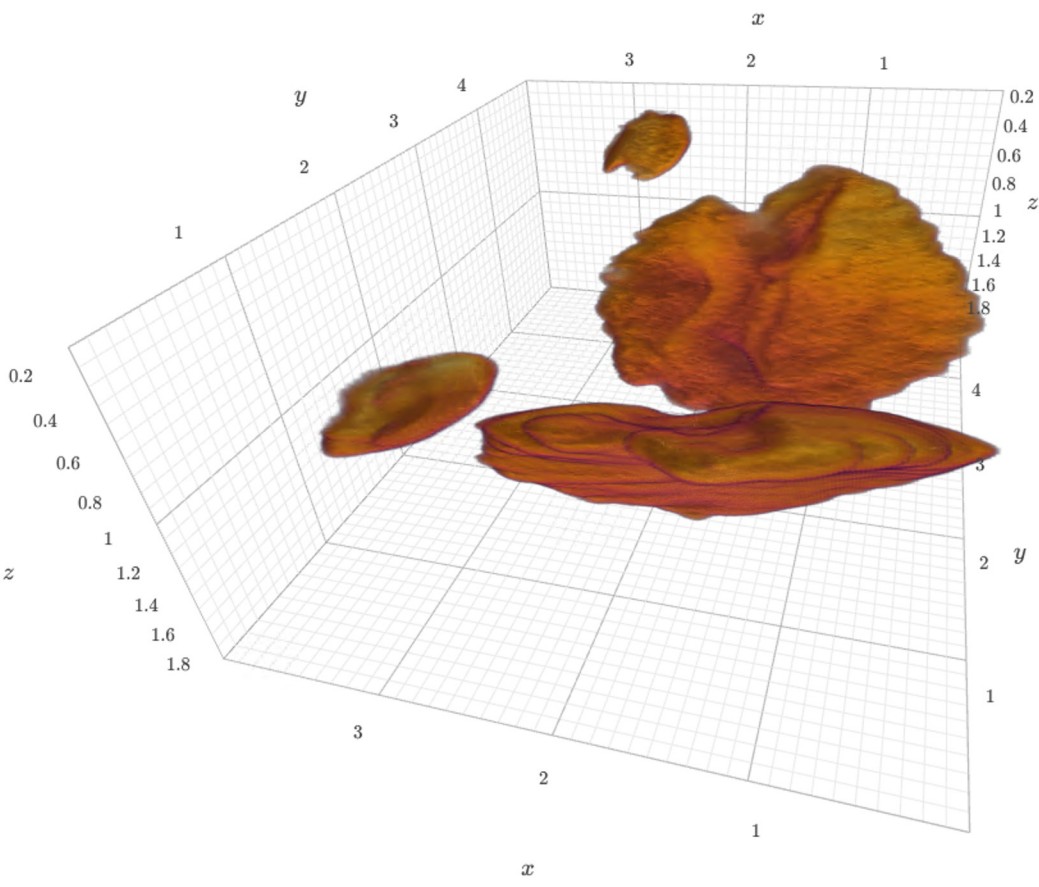

**Fig 3. Three-dimensional view of extracted otoliths of specimen 104016.** The specimen was scanned with an isotropic voxel size of 13 μm. The extracted otolith has a size of approximately 250 x 350 x 150 pixels. The axes labels correspond to mm. A dynamic view of the visualization is available in the Supporting information.

imaged over a wide-spanning range of voxel size (3.5–50 μm) permitting both the analysis of finest details we wanted to resolve (i.e. containing only the oral and pharyngeal jaws) and having datasets (N = 104) containing the whole head for principle components analysis and extraction of the otoliths. In our microCT imaging setup the resulting voxel size is directly correlated with the sample diameter, i.e. smaller specimen were scanned with the smallest voxel size, in the range of several μm. The skulls of larger specimen were scanned with the largest voxel sizes, in the range of several tens of μm. The chosen voxel size is thus specimen- or scan-specific and was always selected such that the desired region of interest fits into the available lateral field of view of the microCT machine at the minimally necessary voxel size to resolve the structures of interest.

## Imaging and preparation for analysis

The whole study we presented here spanned a *long* time frame. It was thus paramount to run the imaging process and the preparation of the tomographic datasets in a batched mode. The *Jupyter* notebook written to prepare the datasets for quality control and analysis were facilitating a short turnaround time for feedback on single scans and such a batched analysis.

## Otolith extraction

The method to extract the otoliths from the tomographic dataset works robustly for all of the different fish sizes and shapes. The extraction is robust because it is based on a combination of distinct details of the gray value curve over the different anatomical directions. The details of the otolith extraction method have been extensively tuned and run in a fully automated way. This allows a highly reproducible and unbiased extraction of the otoliths from the tomographic datasets. This is even the case for one fish which was scanned with a hook still in it's mouth, where the otoliths were nonetheless automatically extracted without any issue.

Data on such automatically extracted otoliths, like volume and geometric information like eccentricity and moments of inertia is biologically interesting as the otoliths grow with the age of the fish. One could help estimate the age of wild fishes using a calibration based on the otolith measurements of a fish of known age. It is worth noting that the estimation of age in tropical fishes are not as simple as for fishes from temperate regions, where one can distinguish summer and winter layers within the otolith [36].

To the best of our knowledge, no fully automated method for non-destructive otolith extraction from high-resolution tomographic data has been published up to now.. Vasconcelos-Filho et al. [37] show a potentially objective method to count growth rings in otoliths of six fish species, but performed their study on extracted otoliths "embedded in a Styrofoam cube", while our method non-destructively extracts the otoliths from the tomographic data. Gu et al. [38]. also provide detailed insight into otoliths and combine microtomographic imaging (with unknown voxel size) with scanning electron microscopy. However, they also mention that the "otoliths were removed [from *Bahaba taipingensis*], cleaned and kept dry for morphologic observation".

## Outlook

The acquired tomographic datasets are the basis for multiple additional analyses of fish morphology.

The presented method offers an insight and algorithm on how to perform tomographic scans, preview and analyze micro-computer tomographic datasets of a large collection of fish. The workflow is relying only on free and open-source software and can thus be used and

verified independently by any interested reader. All the *Jupyter* notebooks described herein are also freely available online [19].

## Supporting information

**S1 Data. Parameters of tomographic scans of all the fishes.** The CSV file ScanningDetails. csv gives a tabular overview of all the (relevant) parameters of all the scans we performed. This file was generated with the data processing notebook and contains the data which is read from *all* the log files of *all* the scans we performed. A copy of each log file is available in a folder in the data processing repository.
(CSV)

**S1 File. Three-dimensional view of *one* of the extracted otoliths.** The three-dimensional view of sample 104016 was generated in the otolith extraction notebook and saved as a self-contained HTML file with *K3D-jupyter*. A copy of this HTML file can be viewed and interacted with through the GitHub HTML preview.
(HTML)

## Acknowledgments

We thank Salome Mwaiko for taking care of the fish collection at Eawag and Mark Charran for helping to find suitable specimens to represent each species. We are grateful to the Microscopy Imaging Center of the University of Bern for the infrastructural support. We thank the manubot project [39] for helping us write this manuscript collaboratively.

## Author Contributions

**Conceptualization:** David Haberthür, Ruslan Hlushchuk.

**Data curation:** David Haberthür, Mikki Law.

**Formal analysis:** David Haberthür.

**Investigation:** David Haberthür, Mikki Law, Kassandra Ford, Marcel Häsler.

**Methodology:** David Haberthür, Mikki Law, Kassandra Ford.

**Project administration:** David Haberthür, Mikki Law, Marcel Häsler.

**Resources:** Marcel Häsler, Ole Seehausen, Ruslan Hlushchuk.

**Software:** David Haberthür.

**Supervision:** Ole Seehausen, Ruslan Hlushchuk.

**Visualization:** David Haberthür, Mikki Law, Kassandra Ford.

**Writing – original draft:** David Haberthür.

**Writing – review & editing:** David Haberthür, Mikki Law, Kassandra Ford, Marcel Häsler, Ole Seehausen, Ruslan Hlushchuk.

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
