## [Decision Letter · Decision Letter 0]

30 Jun 2023

PONE-D-23-10896Microtomographic investigation of a large corpus of cichlidsPLOS ONE

Dear Dr. Haberthür,

Thank you for submitting your manuscript to PLOS ONE. After careful consideration, we feel that it has merit but does not fully meet PLOS ONE’s publication criteria as it currently stands. Therefore, we invite you to submit a revised version of the manuscript that addresses the points raised during the review process.

We look forward to receiving your revised manuscript.

Kind regards,

Michael Schubert

Academic Editor

PLOS ONE

Reviewers' comments:

Reviewer's Responses to Questions

**Comments to the Author**

1. Is the manuscript technically sound, and do the data support the conclusions?

Reviewer #1: Yes

Reviewer #2: Yes

2. Has the statistical analysis been performed appropriately and rigorously? 

Reviewer #1: Yes

Reviewer #2: I Don't Know

3. Have the authors made all data underlying the findings in their manuscript fully available?

Reviewer #1: Yes

Reviewer #2: Yes

4. Is the manuscript presented in an intelligible fashion and written in standard English?

Reviewer #1: Yes

Reviewer #2: No

5. Review Comments to the Author

Reviewer #1: Excellent paper. It was not clear if you could discern daily rings on the otoliths. You might want to discuss if you can use rings on otoliths to age the fish without removing them. Also, do you have to stain the fish to use these rings?

Reviewer #2: I have carefully evaluated the manuscript titled "Microtomographic investigation of a large corpus of cichlids". The authors present a substantial collection of tomographic images from cichlids collected in Lake Victoria, Africa. They showcase significant cranial, maxillary, and otolith images from 362 specimens and provide open-source Python notebooks to facilitate the manipulation and extraction of specific parts. I believe that this work represents a significant contribution to the understanding of cichlids and the application of Micro-CT imaging in fish studies. However, I have some suggestions to improve the manuscript:

The introduction section needs improvement. It is too simplistic and lacks informative details about the state of the art and the justification for the work.

Merging the Materials and Methods section with the Results is not a reasonable way to present the work. It would be better to separate them for clarity.

The section on Micro-computed tomographic imaging should be the core of the paper and explained in more detail. The authors have extensive experience scanning 362 fish specimens of different species and sizes. Therefore, a detailed imaging protocol that can assist readers in obtaining new images should be provided. For example, what criteria did the authors use to select the best set of parameters? It is clear and expected that larger fish would have lower resolution (larger voxel size), but how does the relationship between size and parameters work? How was the filter thickness chosen?

The first sentence of the last paragraph of the subsection Micro-computed tomographic imaging is unclear: "While performing the work, a subset of the data was always present on the production system, for working with it (see Preparation for analysis below)". Please provide more precise information.

The authors should revise the entire "Data analysis" subsection to present a more didactic version that helps any reader use the notebooks. Currently, it seems that only Python users with some expertise can understand what needs to be done. Perhaps the authors could provide a guide or tutorial in the GitHub repository.

How was the sanity check performed (as mentioned in the second paragraph of the subsection "Preparation for Analysis")?

In the same paragraph, the authors mentioned that each tomographic dataset contains around 2700 slices, exceeding the available RAM size on an average high-end workstation. Could you clarify if this limitation affected the analysis and its implications?

The Wikipedia citation in the second paragraph of the subsection "Extraction of oral and pharyngeal jaws..." is not clear. Please provide more specific information or replace the citation with a more appropriate reference.

Regarding the sentence "In total, we compiled an overview of 125 specimens with full head morphology, oral jaw, and lower pharyngeal jaw profiles," why was this done only for a subset of the sample? Please explain the rationale.

The results of the subsection "Principal components analysis of skull landmarks" should be presented in a more accessible manner. Consider providing a tutorial or guide for readers to better understand and apply the analysis. As it currently stands, this subsection may be considered irrelevant for the paper.

The procedure for detecting and extracting otoliths in the subsection "Automatic extraction of otoliths" is not clear. Please provide clearer instructions or guidelines for readers using the notebook.

Use the full words, rather than abbreviations, in figure legends for clarity.

In the discussion section, the authors mentioned acquiring high-resolution tomographic datasets of a large collection of cichlids. However, the statement that "the acquired datasets were imaged over a wide-spanning range of voxel size (3.5–50 μm)" is incorrect. The voxel size is specimen-specific, depending on the size of the fish and the chosen parameters. Please clarify this point to accurately represent the imaging process. The finest resolution was obtained only for small fishes, so specific structures of a fish will have the same image resolution throughout the entire specimen.

It would be beneficial to include references in the discussion section to support the arguments and provide a broader context for the findings.

Overall, the study seems to have spanned a considerable time frame. However, the manuscript's content and structure need improvement to reach a wider audience and effectively communicate the significance of the work. The authors have provided a valuable set of open-source Python notebooks, which is commendable and will be highly useful for the scientific community. However, the workflow needs to be better explained to enable readers to follow the procedures accurately.

By addressing these suggestions, the manuscript will become more comprehensive, accessible, and representative of the valuable contribution made by the authors.

6. PLOS authors have the option to publish the peer review history of their article (what does this mean?). If published, this will include your full peer review and any attached files.

Reviewer #1: **Yes: **Jay Richard Stauffer, Jr.

Reviewer #2: No

---

## [Author Response · Author response to Decision Letter 0]

27 Jul 2023

All 'point-by'-point' responses to the reviewers are in "AnswerLetter.pdf"

---

## [Decision Letter · Decision Letter 1]

21 Aug 2023

Microtomographic investigation of a large corpus of cichlids

PONE-D-23-10896R1

Dear Dr. Haberthür,

We’re pleased to inform you that your manuscript has been judged scientifically suitable for publication and will be formally accepted for publication once it meets all outstanding technical requirements.

Kind regards,

Michael Schubert

Academic Editor

PLOS ONE

Reviewer's Responses to Questions

**Comments to the Author**

1. If the authors have adequately addressed your comments raised in a previous round of review and you feel that this manuscript is now acceptable for publication, you may indicate that here to bypass the “Comments to the Author” section, enter your conflict of interest statement in the “Confidential to Editor” section, and submit your "Accept" recommendation.

Reviewer #1: All comments have been addressed

Reviewer #2: All comments have been addressed

2. Is the manuscript technically sound, and do the data support the conclusions?

Reviewer #1: Yes

Reviewer #2: Yes

3. Has the statistical analysis been performed appropriately and rigorously? 

Reviewer #1: Yes

Reviewer #2: Yes

4. Have the authors made all data underlying the findings in their manuscript fully available?

Reviewer #1: Yes

Reviewer #2: Yes

5. Is the manuscript presented in an intelligible fashion and written in standard English?

Reviewer #1: Yes

Reviewer #2: Yes

6. Review Comments to the Author

Reviewer #1: All modifications to manuscript have been met. I have attached the ms with one minor suggestion. Good job.

Reviewer #2: The authors have significantly improved the manuscript. I also agree with the arguments used to refute some of the suggested modifications.

7. PLOS authors have the option to publish the peer review history of their article (what does this mean?). If published, this will include your full peer review and any attached files.

Reviewer #1: **Yes: **Jay Richard Stauffer, Jr.

Reviewer #2: **Yes: **Paulo J. Duarte-Neto

---

## [Editor Report · Acceptance letter]

18 Sep 2023

PONE-D-23-10896R1 

Microtomographic investigation of a large corpus of cichlids 

Dear Dr. Haberthür:

I'm pleased to inform you that your manuscript has been deemed suitable for publication in PLOS ONE. Congratulations! Your manuscript is now with our production department. 

Kind regards, 

on behalf of

Dr. Michael Schubert 

Academic Editor

PLOS ONE